# Influencing Green Purchase Intention through Eco Labels and User-Generated Content

**Anastasios Panopoulos [1], Athanasios Poulis [2],* , Prokopis Theodoridis [3] and Antonios Kalampakas [4]**

1 Department of Balkan Slavic & Oriental Studies, University of Macedonia, 54636 Thessaloniki, Greece
2 Department of Tourism Management, University of Patras, 26504 Patras, Greece
3 School of Social Sciences, Hellenic Open University, 18 Aristotelous St., 26335 Patras, Greece
4 College of Engineering and Technology, American University of the Middle East, Egaila 54200, Kuwait
* Correspondence: tpoulis@upatras.gr

**Abstract:** The purpose of the study is to investigate how environmental concern, eco-labelling, influencers and user-generated content affect Generation Z's green purchase intention. The objective of this study is to contribute with a new scope that combines influencers and user-generated content on digital platforms with environmental concern for Generation Z. The study also aims to add new value in predicting Generation Z's green purchase intention and results that can be implemented in future marketing strategies. To test the framework, a quantitative research approach, with an online survey, was applied to collect data from Generation Z. The sample size consisted of 393 individuals from Generation Z. Structural Equation Modelling was applied to test the hypothesized framework. All hypotheses were accepted, and hence, this research has identified key variables to predict Generation Z's green purchase intention. Additionally, this paper found that environmental concern has a significant positive impact on Generation Z's user-generated content and eco-labelling, and influencers positively affect Generation Z's user-generated content. This study can aid companies that employ an influencer marketing approach to comprehend how they can motivate customers to buy sustainable products more frequently. This study provides crucial and valuable insights into further understanding how the sustainable consumption behavior of Generation Z can be impacted by the utilization of influencer marketing and their concern for the environment. It also provides a deeper understanding of how influencers and their perceived concerns for the environment can be combined with user-generated content and eco-labelling, as well as subsequent effects on the green purchase intention of members of Generation Z.

**Keywords:** green purchase intention; eco labels; green brand attitudes; user-generated content; influencers; environmental concerns; environmental marketing





## 1. Introduction

Due to increasing consumption, higher carbon dioxide emissions are having a detrimental effect on the environment [1,2]. That has led to a growing consumer concern about the environment as well as to the emergence of new markets for environmentally friendly products. Since consumers' decisions to buy products are now significantly influenced by sustainable consumption [3], they are pursuing buying eco-friendly products that include features that are beneficial for the environment. Although consumers are becoming increasingly more aware of the environment [4,5], and eco-friendly products are beginning to get more and more attention, firms are struggling to predict what drives consumers to purchase eco-friendly goods [6–9]. Therefore, it is critical that marketing professionals comprehend the factors that motivate consumers to purchase eco-friendly products.

Consumers are willing to buy eco-friendly or green products when their purchase reasons are based on environmental benefits. The rationale for green purchasing intention is also described as the propensity of shoppers to purchase certain products based on their

environmental needs [10]. In recent years, a growing body of research has focused on green purchase intentions and behaviors [11–13]. However, there is limited understanding of why consumers have green purchase intention [7,8,14]. Green purchase intention enables individuals to contribute to environmental protection [15]. Joshi & Rahman (2016) state that green purchase intention implies a complex ethical approach to decision-making that is viewed as a type of behavior that demonstrates social responsibility [16]. Examples of eco-friendly consumer behavior include buying products made from recycled materials and recycled organic foods. Other aspects may also include product and packaging design as well as considering environmental aspects in marketing activities such as green advertising and the use of ecolabels [17]. For consumers to determine that products are sustainable and complete the transaction, it is necessary that they have an appropriate eco-label [18]. A definition proposed for the practice of eco-labelling is where "a product claims to furnish consumers with credible and easily accessible information on the environmental attributes of a product" [19].

A key factor in identifying green products is influencer marketing. Influencers have the power to drive sustainable spending behavior, especially among young people. This is of crucial importance given that young consumers have been shown to have a higher affinity with influencers on social media [20]. Additionally, Gen Z members use digital platforms more than anyone else [21] and are exposed to both influencers and user-generated content. In the literature, they are defined not only as the generation that uses social media most frequently, but also as the generation that is most concerned about the environment [21,22]. Pauliene and Sedneva (2019) found that they care about the environment and recognize the importance of sustainable commodities [23]. Due to the fact that Gen Z lacks representation as a consumer group, more research is required to determine whether they are affected by factors that cannot be connected with past research [24]. For the purposes of the current study, Gen Z will be defined as people born in the period from 1995 to 2005 [25].

Thus, the aim of this study is to advance the understanding of how consumers' sustainable spending behavior is influenced by the use of influencer marketing and environmental concerns. Additionally, to the best of the authors' understanding, there is a clear deficiency in the literature in terms of how influencers in combination with consumers environmental concerns could have an impact on the user-generated content and eco-labelling, as well its subsequent effects on the green purchase intention of members of Generation Z. This study will further contribute with a consumer perspective of influencers' role in affecting consumers' sustainable consumption behavior. Regarding the practical implications, this study can aid companies that employ an influencer marketing approach to comprehend how they can motivate customers to buy sustainable products more frequently.

## 2. Theoretical Background and Related Work on Environmental Concerns and Ecolabelling

There is an increased likelihood that consumers whose lifestyles are more ecologically aware will buy green products [26]. Research suggests that consumers are more engaged with eco-labels when their behavior is more environmentally friendly [27]. Consequently, consumers must therefore be positively inclined towards environmental matters for them to increase their involvement with eco-labelled products. Consumer attitudes are significant determinants of which information on green products they will consider and appreciate [28]. Boscolo et al. (2020) argued that consumer behavior is influenced by attitudes via selective attention regarding items that are consistent with this attitude [29]. In this regard, Stone (1984) proposed the concept of attitudinal involvement which suggests that a number of attitudes reflect an individual's fundamental character and motivate him/her to engage with an item [30]. Consumers who are positively inclined towards the purchase of green products become the most involved with them [31]. This demonstrates that the attitudes of consumers regarding whether to buy a green product will have a positive impact on their involvement with eco-labels. As a notion or concept becomes more embedded within the values of consumers, they will begin to exhibit greater involvement with a given

product [32]. In this regard, the extent of this effect is significantly dependent on how knowledgeable they are about the product [33].

Eco-labelling has functioned as a strategic method of communicating products' eco-friendly concerns since organizations have begun to realize how it positively influences the promotion of green products. Furthermore, sustainable production is a concept that has importance from the perspective of firms, whereas sustainable consumption has more relevance to consumers. The mechanism that underpins the influence of eco-labelling was additionally explored with respect to individual difference analysis. The extent of the impact of eco-labelling on sensory ratings is greater in individuals who frequently buy products with eco-labels when purchasing groceries [34] or participate in other environmentally friendly behavior types [8,35]. Hence, environmental concerns may potentially increase the susceptibility of individuals to the effects of eco-labelling [36]. Environmental concern refers to an attitude (i.e., affectively or cognitively evaluating an item) regarding protecting the environment and environmental issues [37,38] and is a factor that influences environmentally friendly preferences and behavior [39].

Various researchers [15,40–42] have recently proposed that environmentally conscious consumption is at least partially reliant on suitable knowledge gleaned from different origins such as advertising, packaging of products (eco-labelling), and different types of programs focused on raising environmental awareness. Moreover, if consumers perceive this information to be trustworthy, they become more reliant on it [43–46]. For instance, Oates et al. (2008) determined that there is an increased likelihood that consumers will buy products from brands based on recommendations from sources in which they have confidence, ignoring information that they perceive to be unreliable [47]. Consequently, when consumers believe that environmental claims are not trustworthy, the probability that they will exhibit pro-environmental behavior will be reduced [48,49].

Digitalisation and the development of social media platforms has offered a new means by which consumers can obtain information regarding reliable sources. Social media channels have facilitated improvements in the communication among diverse stakeholders which has opened the door to engage in dialogues with consumers [20]. Accordingly, Evans et al. (2017) asserted that firms have now increased engagement with consumers via social media by utilizing influencer marketing [50]. In addition to facilitating access to different ways of communicating and engaging with the intended audience, influencer marketing is capable of enhancing the brand awareness of firms via the loyalty of a greater number of new and possible future customers [51]. Individuals with environmental concern who are exposed to green brands via social media could be motivated to purchase eco-friendly items [52]. As a result of the advancement of internet technologies, it is now much easier for consumers to find information to distinguish companies that offer green services or products [14]. This will enable them to focus their green activities to achieve sustainable development and become more competitive in the market.

In their investigation of the green purchase intentions of millennial consumers with respect to their utilization of social media, Bedard and Tolmie (2018) identified that the use of social media is positively correlated with the intention to buy sustainable items. Accordingly, businesses have the ability to motivate consumers in the younger generation to consume sustainably by making them more aware of green products via their social media channels [53]. In this regard, Johnstone and Lindh (2018) contended that "younger generations who lack confidence will allow other determinants to guide their behavioral intentions". Resultantly, it is proposed that social media actors are capable of influencing this consumer segment, particularly, people who are not sufficiently knowledgeable about sustainability. Moreover, Bedard and Tolmie (2018) determined that the purchase intention of millennials with respect to green products has a positive relationship with interpersonal influence on the internet, suggesting that virtual interactions could promote sustainable consumption in this target audience. Therefore, the company's capacity to generate opportunities for consumers to interact like participating in online conversations and

giving feedback could be beneficial by creating a feeling of community, which subsequently enables online interpersonal influence.

### 3. Conceptual Model and Hypothesis Development

*3.1. Environmental Concern and Eco-Labelling*

The growth in environmental awareness and concern in consumers has caused firms to increase their production of more eco-friendly products and development of eco-labels [54,55]. Eco-labelling is a means by which organizations can allow consumers to select a product that has environmental properties with minimal involvement. Therefore, the definition given by Sønderskov & Daugbjerg (2011) is appropriate, as they claimed that an eco-label is "a product claim to furnish consumers with credible and easily accessible information on the environmental attributes of a product".

Efforts to authorize and produce eco-labelling were pioneered by the World Wide Fund for Nature (WWF) through the establishment of the professional buyer's network in 1991 [18]. This facilitated the production and promotion of products with eco-labels by retailers which resulted in an expansion in the volume of such products. Additionally, eco-label certification eliminates the information disparity and provides assurances to consumers that the products conform with their environmental concerns [41]. There has been significant growth in the market for green products, and consumers now prefer greener options that can lower their carbon footprint [4]. Thøgersen (2000) stated that for consumers to have the ability to identify an eco-labelled product, it is necessary for them to have existing knowledge or concern for the issue. Hoogland et al. (2007) further affirmed that values like protection of the environment as well as concern for health and the environment are related to eco-labels. Hence, consumers who have environmental concern exhibit indications to influence eco-labels.

As well as having environmental concern [56,57], members of Generation Z are characterized as being lighthouse customers, meaning that there is an increased likelihood that they will experiment with new products and can more easily understand eco-labelling [21,26]. Moreover, Song et al. (2020) claimed that most members of Generation Z are environmentally aware and have concern for the environment, with more than 50% of respondents expressing the desire for firms to offer an increased number of eco-labelled products. Accordingly, the study presents hypothesis H1a:

**H1a.** *The environmental concerns have a positive impact on the use of eco-labels by firms.*

*3.2. Environmental Concern and User-Generated Content*

In recent years, there has been consistent expansion in social media, while the average user age is continuously decreasing [58]. Social media networks are employed for obtaining and sharing information on trending issues that have particular importance for the younger generation [24]. Thus, it is important to explore how content generated by users from Generation Z is influenced by environmental concern which is an issue that they are confronted with on a daily basis [21].

Sparks et al. (2013) noted that consumers who have environmental concerns are more likely to focus on, share, and seek information regarding environmental matters via social media channels [59]. This conforms with previous studies, suggesting that consumers who possess existing knowledge on a subject search for and share knowledge to acquire more outside information [60,61]. Therefore, exhibiting environmental concern could drive consumers to share their own knowledge and search for content generated by other uses on environmental concerns. Moreover, it has been demonstrated that social media platforms can influence the participation of individuals in eco-friendly activities [62]. Therefore, this indicates that consumers with environmental concerns could produce user-generated content that promotes environmental issues.

Even though mass media is a critical means of promoting awareness on environmental matters, consumers from younger generations are increasingly using different communica-

tion channels for the purpose of sharing and addressing issues [63]. Social media networks provide consumers with the chance to form communities in which needs are addressed more specifically [64]. Modern consumers from the younger generation who have similar environmental concerns utilize social media networks for the purpose of creating groups in which they have the opportunity to actively generate and disseminate user-generated content regarding environmental problems in text and video format [63]. Additionally, researchers have argued that members of Generation Z generally utilize social media platforms for sharing and seeking information on important and consequential issues [25,65]. Hence, hypothesis H1b is proposed:

**H1b.** *The environmental concerns have a positive impact in the creation of User-Generated Content (UGC).*

### 3.3. Influencers and Eco Labelling

Multiple studies have previously reviewed the positive attitudes of consumers regarding green issues in the Western context [66–68]. Connelly (2011) described the importance of consumer preferences for environmentally friendly brands [69], emphasizing the significance of all companies recognizing such choices and developing strategic plans to integrate them into their business models [70]. It is necessary for companies to adopt more sustainable approaches and focus on customers in their sustainability marketing activities (Lim, 2017) [71]. Via social media channels, the reliability of communication with consumers has increased, thus molding their purchasing intentions [72]. Consumers alter how product and service information is accessed, and classical shopping methods have experienced a rapid evolution [73]. Aprile and Fiorillo (2017) contended that exposure via the media significantly enhances the extent to which important environmental concerns are disseminated and the intended audiences become environmentally aware [74]. Influencers have the ability to shape and increase awareness regarding environmental matters and green consumption behavior [75]. Furthermore, how influencers behave on social media could affect the attitudes, mindsets, views, intentions, and behavior of those who follow them, which is largely attributable to the content they share [76].

The roles of influencers have gradually evolved from merely posting unofficial information to making full use of the potential offered by the connectivity of social media [77]. Analysis of the strength of opinion leaders' voices, with respect to the effect of their user-generated content, reveals that influencers are capable of influencing the eco-labelling process due to the fact that it is a preferred information instrument for consumers with environmental awareness which shapes their decision to buy [78]. Following these concerns, it is hypothesized that influencers significantly affect eco-labelling.

**H2a.** *Influencers have a positive impact on the use of eco-labels by firms.*

### 3.4. Influencers and User-Generated Content

People exhibit sustainable engagement through the promotion of eco-friendly products and the encouragement of green behavior via social media channels [79]. The aim is for sustainable products to be normalized, which can be achieved by influencers by reinforcing their assertions with evidence in addition to maintaining transparency and explaining how sustainable lifestyles can be beneficial. It is also possible for sustainable influencers to deal with extant concerns pertaining to sustainable products to make the products more credible. Chwialkowska (2019) further elucidated that influencers who communicate regarding sustainability tend to concentrate on the advantages of green behavior for individuals rather than the tangible environmental impacts of sustainable consumption. Therefore, sustainable communication of influencers is associated with content that entails important information. Accordingly, Lou and Yuan (2019) contended that the informative value of influencers will positively affect the purchase intentions of those who follow them [80].

Influencers are considered to be people who engage in opinion leadership on social media networks and operate as a reference group for members of Generation Z [81]. Additionally, they have previously been called "micro-celebrities" in the literature [82–84]. They utilize social media as their main means of communicating and it is inevitable that they will significantly affect people who use such platforms [85–87]. Research has shown that members of Generation Z use their digital devices for more than 10 h per day, where in excess of two-thirds of that time is devoted to watching videos and using social networks like YouTube, Twitter, Instagram, TikTok, Snapchat, and Facebook [88]. The extent to which influencers impact members of Generation Z and are subjected to the narratives of influencers suggests that they are engaged with them and incorporate them into their user-generated content [89]. Both Jin & Puha (2014) and Rifon et al. (2016) suggested that influencers can shape the content generated by consumers as well as their eWOM [90,91]. These researchers concurred that influencers have the ability to increase the engagement of consumers via social media.

Additionally, influencers are capable of creating discussions and raising awareness regarding particular subjects using hashtags which influence user-generated content [92,93]. Consequently, the influencer's narrative, namely, environmental consciousness, will be augmented via user-generated content using hashtags. This was reinforced by Keller and Berry (2003) in earlier influencer theory in which they claimed that leaders of opinion can influence people's views and behavior. This effect is exemplified by the impact that Greta Thunberg has had on consumers, particularly those from Generation Z which has been defined in the literature as the "Greta effect" [94,95]. Thunberg impacted people using social media channels, and she was frequently referenced by users in the content they generated [95,96]. Researchers have argued that a causal relationship exists, whereby the user-generated content of Generation Z is affected by influencers [89,93,95,97]. Therefore, H2b is suggested:

**H2b.** *Influencers have a positive impact on the creation of User Generated Content (UGC).*

### 3.5. Eco-Labelling and Green Purchase Intention

Eco-labelling enables consumers to be stimulated to select a green product with minimal involvement. Furthermore, eco-labelling makes consumers aware of the impact they will have on the environment if they buy the product [98]. Grankvist et al. (2004) claimed that if retailers use eco-labels on their products, they will aid consumers with making their green purchases, and this commitment is derived from the environmental consciousness and concern of consumers [99]. This was reinforced by Teisl et al. (2017) who investigated the responses of consumers to eco-labelling [100]. They offered evidence based on the market that people are positively inclined towards eco-labelling as well as the fact that the use of eco-labels leads to growth of the sales of a specific product. Moreover, the green characteristics promoted by the eco-label significantly increase the likelihood that consumers will buy the product assuming that they have awareness of eco-labels [52,101,102]. Therefore, the use of eco-labels is an exact means of guiding the decisions of consumers and promotes the commitment of consumers to their green purchases.

With respect to Generation Z, Grankvist et al. (2004) claimed that it is assumed that they have positive attitudes towards products with eco-labels, thus suggesting that such labels will impact their green purchase intention. Song et al. (2020) agreed with this finding, confirming that the green purchase behavior of younger people is affected by eco-labels according to the consumption of eco-friendly products. Additionally, Fiala et al. (2016) determined that the use of eco-labels impacts the behavior of Generation Z [103]. Hence, hypothesis H3 is presented:

**H3.** *The eco-labels used by firms by firms have a positive impact on the green purchase intention.*

### 3.6. User-Generated Content and Green Purchase Intention

User-generated content began to increase parallel to the rise of digital platforms. Dichter (1966) conducted a ground-breaking study on how people share information about products and according to his study, consumers expressed an interest in offering their opinions to other people, defined as "word of mouth" [104]. Via digital platforms, user-generated content acts as a medium through which information can be disseminated and awareness raised regarding a particular issue [105]. Consumers from Generation Z can observe other people sharing information they probably would not have encountered in the absence of user-generated content. Moreover, it has been determined that members of Generation Z are influenced by the user-generated content of their peers which motivates them to make the purchase [106–108]. Reviews generated by users are considered to be content with the highest level of trustworthiness, and Generation Z seek them when searching for information on the internet. Furthermore, Sethna et al. (2017) identified that user-generated reviews are employed by 81% of consumers when deciding whether to make a purchase [109]. Therefore, user-generated content is a critical factor in the process followed by consumers prior to making a purchase.

With respect to green purchase intention, consumers generally utilize social media networks to gather knowledge, which is a mechanism in the process of committing to a purchase [106]. Ransbotham et al. (2012) stated that social media networks that include user-generated content comprise environments could strengthen green behavior intention, including those regarding green purchases [110]. Moreover, user-generated content that assumes viral status along with the identified causation of such content has been determined to have an increased effect on purchase intention [67]. Members of Generation Z, who have the ability to acquire information rapidly according to the user-generated content of consumers, have access to viral content, and therefore may have increased susceptibility. Even though the significance of user-generated content and the manner in which it is utilized by consumers in their decision-making processes regarding commitment to purchase has been supported in past studies [111,112], it must be recognized that user-generated content and reviews are not used by every consumer. Hence, the current study presents hypothesis H4:

**H4.** *User-generated content has a positive impact on green purchase intention.*

By synthesizing all the above this study proposes the following framework (Figure 1).

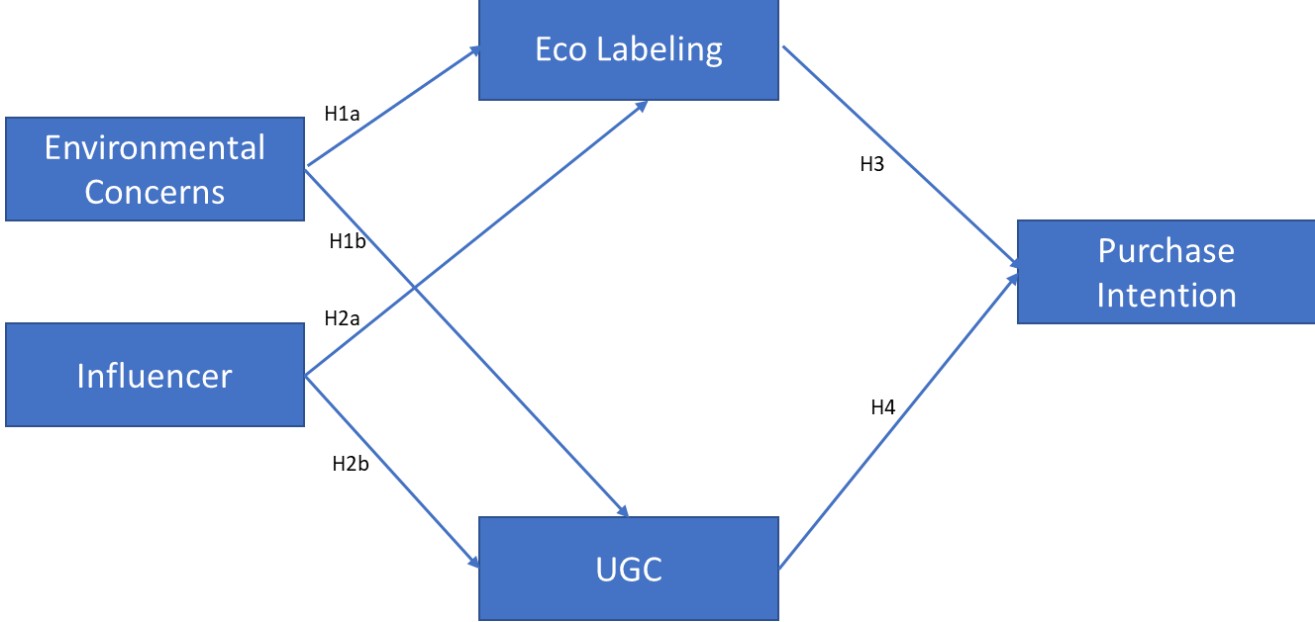

**Figure 1.** Proposed framework.

### 4. Methodology

For the creation and validation of the questionnaire, a qualitative exploratory research was carried out based on the literature review outlined in this paper's theoretical framework. A small focus group was held with four marketing communication academics and four social media communication practitioners, as participants. These experts were carefully selected based on their level of competence in this study. Their responses were examined using a content analysis method, which according to Churchill (1979) is a well-established qualitative data analysis tool [113]. This approach is generally employed when creating construct domains, evaluating the conceptual model's nomological validity, and looking at both current and modified norms. Based on the findings of the analysis, a preliminary questionnaire was created and then was pretested with 9 marketing managers experienced in FMC to ensure content validity [114]. This process led to the rephrasing of 9 items, based on their suggestions.

The questionnaire of the study was then distributed electronically, which was likely to increase the number of participants and improve the representativeness of the target consumer group [115]. Furthermore, the questionnaire was conducted electronically to ensure survey quality and eliminate any human error at the data entry level [116,117]. By creating awareness and providing secure access, any potential bias in coverage is minimized [118]. Following Armstrong and Overton (1977), we also adopted the time-trend procedure to identify any differences between early and late respondents [119]. No differences were found; hence, non-response bias does not appear to be an issue in the current study.

The respondents, after reading the consent form, had to answer a filter question regarding the age group (Gen Z). We purposefully used this filter question to reduce respondent burden and not to collect meaningless data. The target population chosen for this study were students in the UK, aged between 18 and 25. According to a survey of social media users aged in that age group in the UK, 71 percent of respondents use the Instagram mobile app daily. Approximately 70% of the respondents reported using social video app TikTok and YouTube daily, respectively. When considering the levels of participation in social media usage in the UK, a trend is clearly identifiable. Nearly half of Gen Z (46%) claims that, when faced with well-executed branded content, they feel inspired to make a purchase, and a quarter (24%) claims that they would share it with their friends. At the same time, a fifth (21%) claims that would subscribe to a brand's email database. Apparently, that has a huge impact on firms, since the specific market segment will be a regular consumer, with more buying power, in the near future. Hence, firms need to create more meaningful and appealing content.

A total of 419 responses were gathered; however, only 393 were usable, out of a targeted 914 participants, from 5 different UK universities, thus achieving a satisfactory response rate of 42.9%. These students were sent an email invitation with a short description of the study, information about confidentiality and a link to the survey. After two weeks, a reminder email was sent to those who had not responded and after another week, they received a final reminder. The survey was hosted on Qualtrics, an online survey hosting site, and was fielded in March to May 2021. A screening question regarding their age was used to make sure that the participants, although students, fit the criterion of the specific age bracket (18–24).

A total of 209 (57.8%) were female, while 166 (42.2%) were male. For ethnicity, 194 (49.36%) were White, 92 (23.41%) were African American, 84 (21.37%) were Asian, 14 (3.56%) were Hispanic, and 9 (2.29%) were Other. In terms of their year in the university, 90 (22.9%) were 1st, 102 (25.95%) were 2nd, 92 (23.41%) were 3rd year undergrad, and 110 (27.99%) were postgraduate students.

Within survey design, operationalization is a crucial process; it transforms theory into research measurements [115]. The measurements were applied from previous relevant literature to ensure validity and reliability. We examined the validity and reliability of the data (Table 1) for the purpose of ensuring the suitability of the scales, all of which were

above the minimum thresholds [120]. Regarding the measurement, a 5-point Likert scale was applied, where the scales vary from 1 (strongly disagree) to 5 (strongly agree). Table 2 displays the operationalization where definition, items used, and adoption is implemented to ensure a valid and reliable questionnaire.

**Table 1.** Reliability and validity of the construct variables.

| Constructs | Items | M | SD | AVE | CR | Cronbach a |
|---|---|---|---|---|---|---|
| Green Purchase Intention | GPI1 | 3.66 | 0.782 | 0.564 | 0.864 | 0.807 |
| | GPI2 | 3.78 | 0.840 | | | |
| | GPI3 | 3.07 | 1.127 | | | |
| | GPI4 | 4.03 | 0.857 | | | |
| | GPI5 | 3.92 | 0.811 | | | |
| Environmental Concerns | EC1 | 4.46 | 0.642 | 0.508 | 0.861 | 0.809 |
| | EC2 | 4.27 | 0.729 | | | |
| | EC3 | 4.23 | 0.732 | | | |
| | EC4 | 4.20 | 0.750 | | | |
| | EC5 | 4.19 | 0.808 | | | |
| | EC6 | 4.31 | 0.722 | | | |
| Eco-label | EL1 | 3.70 | 0.747 | 0.580 | 0.892 | 0.855 |
| | EL2 | 3.65 | 0.723 | | | |
| | EL3 | 3.65 | 0.836 | | | |
| | EL4 | 3.58 | 0.798 | | | |
| | EL5 | 3.70 | 0.747 | | | |
| | EL6 | 3.11 | 1.141 | | | |
| User- generated content | UGC1 | 4.16 | 0.795 | 0.569 | 0.828 | 0.753 |
| | UGC2 | 4.08 | 0.747 | | | |
| | UGC3 | 4.16 | 0.737 | | | |
| | UGC4 | 3.49 | 1.143 | | | |
| | UGC5 | 3.59 | 1.285 | | | |
| | UGC6 | 3.81 | 0.859 | | | |
| Influencers | I1 | 3.49 | 1.055 | 0.595 | 0.910 | 0.881 |
| | I2 | 3.42 | 1.022 | | | |
| | I3 | 3.44 | 1.047 | | | |
| | I4 | 3.27 | 1.177 | | | |
| | I5 | 3.85 | 0.950 | | | |
| | I6 | 2.97 | 1.186 | | | |
| | I7 | 3.83 | 0.926 | | | |

**Table 2.** Definitions and items of the construct variables.

| Green Purchase Intention | | |
|---|---|---|
| The tendency of a buyer to purchase a specific product based on the environmental necessity [10]. | GPI1: I choose to buy products that are environment friendly. GPI2: I intend to buy green products next time because of its positive environmental contribution. GPI3: I buy green products even if they are more expensive than the non-green ones. GPI4: I prefer green products over non-green products when their product qualities are similar. GPI5: I often buy products that use recycled/recyclable packaging. | [121] |
| Environmental Concerns | | |
| The degree to which people are aware of problems regarding the environment and support efforts to solve them and or indicate the willingness to contribute personally to their solution [122]. | EC1: I am concerned about the current environmental state the world is in. EC2: When humans interfere with nature, it will cause serious consequences. EC3: The balance of nature is very delicate and can be easily upset. EC4: I am willing to reduce my consumption to help protect the environment. EC5: Modern development threatens the environment. EC6: The effects of pollution on public health are worse than we realise. | [121] |

**Table 2.** *Cont.*

| Green Purchase Intention | | |
| --- | --- | --- |
| **Eco-*label*** | | |
| A product claim to furnish consumers with credible and easily accessible information on the environmental attributes of a product [18]. | EL1: Products endorsed by eco-labels are credible. EL2: Products endorsed by eco-labels comply with quality environmental standards. EL3: Eco-labels are a reliable source of information about the environmental quality and performance of a product. EL4: Most of what eco-labels say about products is true. EL5: Eco-labels inform consumers about the environmental safety of a product. EL6: I search for any logo or label on the product endorsing environmental concern when buying any product. | [36] |
| **User-generated content** | | |
| Encompassing opinions, experiences, advice and commentary about products, brands, companies, and services—usually informed by personal experience—that exist in consumer-created postings on internet discussion boards, forums, Usenet newsgroups and blogs [123]. | UGC1: I believe user comments/reviews of a product are more beneficial than manufacturer provided information. UGC2: I trust user comments/reviews of a product to be reasonably accurate representations of a product. UGC3: I trust reviews from friends or people I follow on social networking websites. UGC4: I would trust a product review posted by an average user more than a product review posted by an expert. UGC5: I have written a comment/review for a product, brand, or personality on an online platform. UGC6: I generally find it easy to exchange ideas with participants on digital platforms. | [109] |
| **Influencers** | | |
| Opinion leaders on social media platforms who have the ability to influence other individuals [124]. | I1: The influencers I follow remind me of someone who is competent and knows what he/she is doing. I2: The influencers I follow have the ability to deliver what they promise. I3: The influencers I follow have believable claims. I4: The influencers I follow do not pretend to be something they are not. I5: I think an advertisement with an influencer who has expertise (skilled, qualified, knowledgeable, experienced) is more respectable. I6: I think a brand being endorsed by influencers is more trustable. I7: Influencers help me to remember a brand or a product. | [125] |

## 5. Findings and Analysis

The results indicated a good model fit (Cmin/df = 2.79; CFI = 0.858; NFI = 0.798; RMSEA = 0.068). All standardized coefficients were significant (ranging from 0.238 to 0.863). Following Fornell and Larcker's (1981) criteria, discriminant validity was assessed by comparing the square root of AVE with the correlations of constructs. Table 3 shows that the square roots of the AVE of each construct was greater than the correlation coefficients between constructs. Thus, discriminant validity for the present model was confirmed.

**Table 3.** Discriminant Validity of the construct variables.

| | EC | I | UGC | EL | GPI |
| --- | --- | --- | --- | --- | --- |
| Environmental Concerns | 0.800 * | | | | |
| Influencers | - | 0.904 * | | | |
| User-generated content | 0.407 | 0.414 | 0.718 * | | |
| Eco-label | 0.286 | 0.492 | 0.320 | 0.870 * | |
| Green Purchase Behavior | 0.225 | 0.318 | 0.381 | 0.522 | 0.822 * |

EC: Environmental Concerns, I: Influencers, UGC: User-generated content, EL: Eco-label, GPI: Green purchase intention. * Diagonal: Square root of AVE.

Further, it is crucial to examine composite reliabilities of the construct [126]. Composite reliability refers to a principal measure in assessing measurement models [126,127] (Hair, 2011; Fornell and Larker, 1981) indicating the degree of the internal consistency of the construct indicators [127]. The composite reliabilities and the values of average variance extracted are presented in Table 4. The construct shows evidence of composite reliability.

**Table 4.** Composite reliabilities and AVE of the construct variables.

|  | EC | I | UGC | EL | GPI |
|---|---|---|---|---|---|
| Composite reliability | 0.810 | 0.887 | 0.754 | 0.858 | 0.810 |
| AVE | 0.640 | 0.817 | 0.516 | 0.757 | 0.675 |

EC: Environmental Concerns, I: Influencers, UGC: User-generated content, EL: Eco-label, GPI: Green purchase intention.

A summary of the statistics related to the estimations and tests of the hypotheses is presented in Table 5. As depicted by the table, Environmental Concerns demonstrated a positive impact on Eco-Labelling, which confirmed hypothesis H1a ($\beta = 0.286$; $p$-Value = 0.000). The findings are in accordance with Hoogland et al. (2007) and Thøgersen (2000), where the effect of environmental concern on eco-labelling is established [26,128]. Environmental Concerns demonstrated a positive effect on User-Generated Content and thus H1b is accepted ($\beta = 0.407$; $p$-Value = 0.000). The studies of Gursoy & McCleary, (2004) and Rao & Sieben, (1992), who found that consumers who are environmentally concerned share their knowledge and concern with others, support the results. Furthermore, the results align with Amandeep & Chahal (2018), who argued that younger consumers tend to create and share content on digital platforms.

**Table 5.** Hypotheses testing and standardized structural coefficients.

| Hypothesis | β * | s.e. | c.r. | Acceptance |
|---|---|---|---|---|
| H1a Environmental Concerns → Eco Labelling | 0.29 | 0.058 | 5.08 | Supported |
| H1b Environmental Concerns → User-Generated Content | 0.41 | 0.063 | 6.007 | Supported |
| H2a Influencer → Eco Labelling | 0.49 | 0.034 | 8.783 | Supported |
| H2b Influencer → User-Generated Content | 0.41 | 0.034 | 6.756 | Supported |
| H3 Eco Labelling → Purchase Intention | 0.45 | 0.057 | 6.435 | Supported |
| H4 User Generated Content → Purchase Intention | 0.24 | 0.058 | 3.737 | Supported |

* = standardized values; $p < 0.001$; Note: Cmin/df = 2.79; CFI = 0.858; NFI = 0.798; RMSEA = 0.068.

Influencers positively impact eco-labelling; and therefore, H2a was also accepted ($\beta = 0.492$; $p$-Value = 0.000). The outcomes concur with those of Bedard and Tolmie (2018) who emphasized that the expansion of social media has led to novel and creative methods of conveying information about the sustainability of products, whereby the engagement and cooperation of consumers in the purchasing process are enhanced. Due to the fact that consumers often resort to social media to obtain information regarding products, the extent to which a firm is capable of adapting to emerging calls for environmentally friendly products will have a significant effect on the decision to purchase. With respect to H2b, as shown in the table, Influencers positively affect User-Generated Content ($\beta = 0.414$; $p$-Value = 0.000). This implies that opinion leaders on online platforms influence Generation Z users with regard to their postings and comments on online platforms or forums. According to Keller and Berry (2003), Kim and Seo (2020), and Kim et al. (2019), this is because influencers have the power to raise awareness regarding a specific subject, and due to their position as opinion leaders, they are capable of creating a narrative which influences consumers.

Regarding H3, as depicted by Table 3, Eco-Labelling has a positive effect on Purchase Intention ($\beta = 0.446$; $p$-Value = 0.000). The results correspond well with the conclusion of Grankvist et al. (2004); that younger generations have positive associations with eco-labelling. Moreover, the study by Matthes et al. (2014) and Zhao et al. (2014) pointed out that having eco-labels on products guides consumers' decision-making process and makes it easier for consumers to commit to a green purchase [129]. In addition to the above, H4 was accepted ($\beta = 0.238$; $p$-Value = 0.000) through the conclusion that User-Generated Content has a positive effect on Purchase Intention. This is similar to the results of Abdul et al. (2015). The results indicate that Generation Z utilizes user-generated content in its reasoning to commit to green purchases (Figure 2).

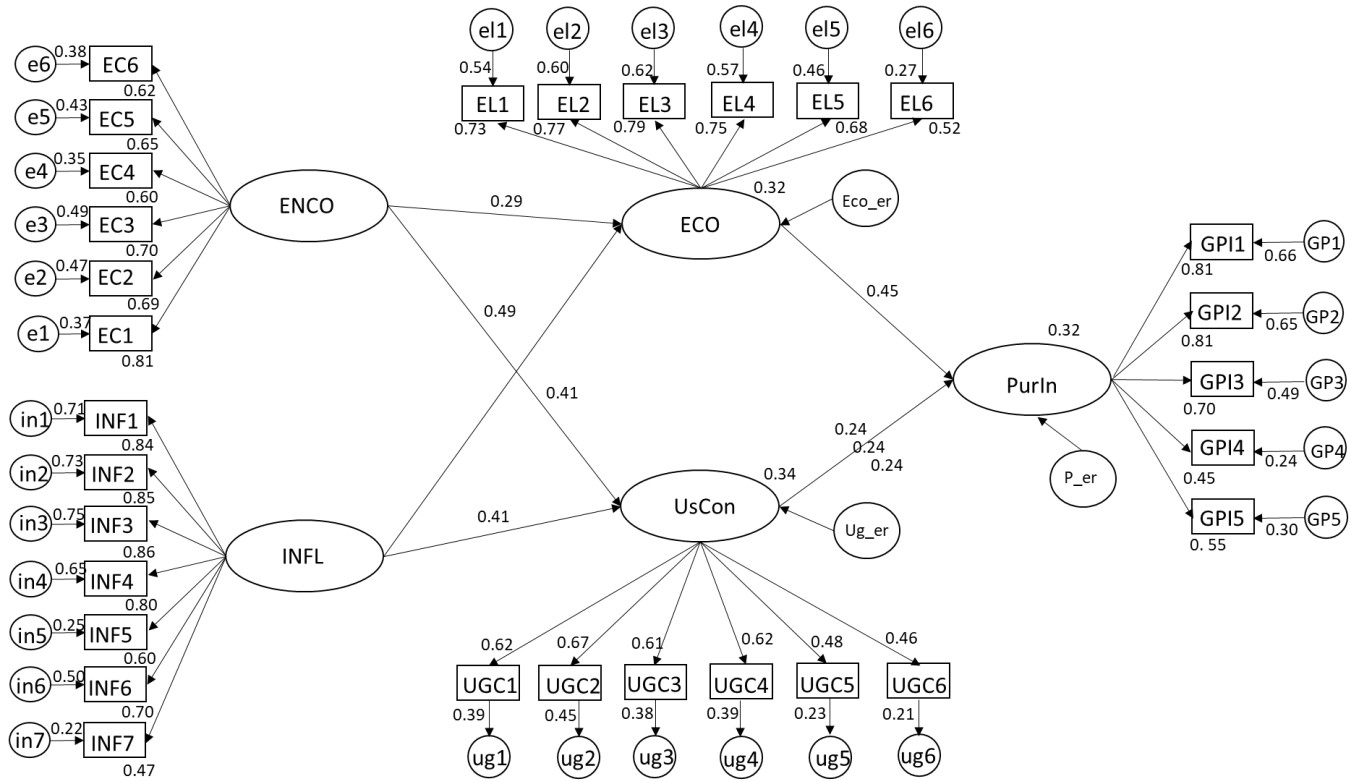

**Figure 2.** Structural Equation Modelling output.

## 6. Discussion and Managerial Implications

The findings provide important insights into understanding Generation Z's green purchase intentions and their user-generated content, which can inform marketing approaches for the specific group. The analysis in this study showed that the proposed framework is statistically significant and thus supports all hypotheses.

This study suggests that both influencers and consumers' environmental concerns positively influence both environmental labelling and user-generated content. These findings are useful indications for marketers in developing strategies. Firms should take into consideration the importance of eco-labels for the consumers and also comprehend the way influencers affect Gen Z green purchase intention. The findings can be used as an insight when marketing managers strategize their online advertisement on online platforms that target the specific generation. With consumers becoming more environmentally conscious, brands—especially those that cite sustainability as one of their core values—are being forced to rethink their marketing strategies. Brands now need to be more innovative to attract the attention of influencers by creating a sense of excitement to share while advocating for sustainability.

The results suggest that influencers are at the forefront of the sustainability debate, acting as role models, promoting calm and healthy lifestyles, and supporting social change across a range of critical issues. Brands should try to appeal to the influencers who are focusing on sustainability due to several factors. First, the emerging phenomenon defined as "cause marketing", where brands connect their image to important issues, is an important way to engage Gen Z. In today's digital world, most discussions of important issues take place online, and such discussions are dominated by influencers. Brands gain the ability to communicate messages that inspire consumers by developing richer and more satisfying marketing campaigns.

Additionally, eco-labeled products are a commercial success due to the positive public image that can be created by influencers who may persuade consumers to purchase green products while increasing brand loyalty. The results also suggest that influencer-generated

content can have a positive impact in the motivation of consumers to purchase green products. Consumers believe that such content improves their knowledge of green products and supports their decision-making process.

The current research has attempted to contribute in the fields of ecolabelling, environmental issues, and green purchasing, by providing scholars with a deeper insight on this relationship. Firstly, this research provided statistical verification of the effect that influencers have on eco-labelling as well as the manner in which green purchase behavior is significantly impacted by eco-labelling. Additionally, it demonstrated the process and mechanism with respect to how environmental concerns could impact the green purchase behavior of consumers from a statistical perspective.

## 7. Limitations and Future Research

The focus of this study on Gen Z members aged 18 years and older may limit generalization to wider populations. Therefore, future studies may be conducted to further analyze the framework when all members of Gen Z are 18 years of age or older.

**Author Contributions:** Writing—original draft, A.P. (Anastasios Panopoulos), A.P. (Athanasios Poulis), P.T. and A.K.; Writing—review & editing, A.P. (Anastasios Panopoulos), A.P. (Athanasios Poulis), P.T. and A.K. All authors have read and agreed to the published version of the manuscript.

**Funding:** This research received no external funding.

**Institutional Review Board Statement:** Not applicable.

**Informed Consent Statement:** Informed consent was obtained from all subjects involved in the study.

**Data Availability Statement:** Data is unavailable due to privacy.

**Conflicts of Interest:** The authors declare no conflict of interest.

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
