# Peer review of "Influencing Green Purchase Intention through Eco Labels and User-Generated Content"

_sustainability, doi:10.3390/su15010764_

Round 1
Reviewer 1 Report (Previous Reviewer 2)
Now the paper is improved.
Author Response
We would like to thank the reviewer for the valuable feedback that strengthen the paper.
Reviewer 2 Report (Previous Reviewer 3)
As I said in my previous review, this is an interesting study that explores the purchase intentions of those labelled “Gen Z” consumers, with a particular focus on their environmental concerns. The study appears to have been carried out diligently and presents useful findings.
I would like to thank the authors for amending their manuscript in light of my previous comments. I particularly appreciated the way in which the contribution of the paper is argued, and the clearer description of the methodology.
Author Response
We would like to thank the reviewer for the valuable feedback that strengthen the paper.
Reviewer 3 Report (New Reviewer)
There are several inputs that can be used as improvements in the text of this article:
> It is necessary to emphasize the problems in the introductory section to bring up GAP Research by maximizing the use of empirical trends and theoretical GAPs.
> In the method that requires additional sources, it is used (Hair et al., 1998), even though Hair has the latest source books, but there are no references in the bibliography.
> There are still many citations that do not have bibliographies, and there are also bibliographies whose citations are unclear, this indicates a violation of ethics and plagiarism in writing scientific articles
> The results section needs to display the results of the proposed model fit test and maximize the results (Goodness of Fit) with a minimum standard stating the suitability of the research model
> Conducting in-depth discussions by confirming and interpreting data findings through previous research, concepts and also theories related to the variables studied.
> Conclusion: complete with answers to research problems, recommendations, implications, limitations and future research.

Author Response
It is necessary to emphasize the problems in the introductory section to bring up GAP Research by maximizing the use of empirical trends and theoretical GAPs.
Response: We would like to thank the reviewer for the comment. We had rewritten the introduction in order to meet the requirements of the previous reviewers. We believe that the new introduction clearly states the gaps and the necessity of this research.
> In the method that requires additional sources, it is used (Hair et al., 1998), even though Hair has the latest source books, but there are no references in the bibliography.
Response: We would like to thank the reviewer for the comment. We agree and we have amended the sources.
> There are still many citations that do not have bibliographies, and there are also bibliographies whose citations are unclear, this indicates a violation of ethics and plagiarism in writing scientific articles
Response: We would like to thank the reviewer for the comment. We agree and we have amended the intext as well as the final list of references.
> The results section needs to display the results of the proposed model fit test and maximize the results (Goodness of Fit) with a minimum standard stating the suitability of the research model
Response: We would like to thank the reviewer for the comment. We have moved the following in the Analysis section. The results indicated a good model fit (Cmin/df = 2.79; CFI = .858; NFI = .798; RMSEA = .068). All standardized coefficients were significant (ranging from 0.238 to 0.863). Following Fornell and Larcker’s (1981) criteria, discriminant validity was assessed by comparing the square root of AVE with the correlations of constructs. Table 3 shows that the square roots of the AVE of each construct was greater than the correlation coefficients between constructs. Thus, discriminant validity for the present model was confirmed.
> Conducting in-depth discussions by confirming and interpreting data findings through previous research, concepts and also theories related to the variables studied.
Response: We would like to thank the reviewer for the comment. We have tried to underpin any relevant literature that verifies our results in the Findings and Analysis section.
> Conclusion: complete with answers to research problems, recommendations, implications, limitations and future research.
Response: We would like to thank the reviewer for the comment. We have tried to provide recommendation and implications in the Managerial implications section. We have now added limitation and future research as per your recommendation.
Round 2
Reviewer 3 Report (New Reviewer)
Thank you for revising the results of the previous review, it's just that there are a few things the author hasn't done to improve. Therefore I immediately provide comments and also improvements that should be on the part that needs to be repaired or perfected. Please correct as much as possible and see directly the parts of the manuscript that are very important and basic, so as not to experience a violation of writing ethics.

This manuscript is a resubmission of an earlier submission. The following is a list of the peer review reports and author responses from that submission.
Round 1
Reviewer 1 Report
The conceptual framework presented by the manuscript is good and theoretically viable.
However, the paper is seriously flawed based on the following observations:
1. The topic is very broad and the approach to it is also very broad. As it is, this is actually good for introductions to the subject matter in a book or magazine, but not in scientific journal articles.
2. The statistical approach is theoretically sound but is not reliable and valid as applied in the study. There is no definition of the scope of the study, target area where the study is conducted, tested, and validated. There are millions of people belonging to Generation Z, but the sample size is just a very negligible number (419 responses); thus, the observations claimed are not reliable and valid for all people belonging to Generation Z in the whole world. Thus, generalisations, not even the observations based only on 419 responses ran through some statistical tools, cannot be applied to all Generation Z people.
3. There is a need to define and conduct profiling of the demographics of the target population considered in the study who belong to Generation Z. Although the study involves online and social media interactions, people belonging to Generation Z may have varied choices or preferences based on their background, educational, social and financial status, culture, and so on. This is not covered nor discussed thoroughly in the paper and how these aspects can or cannot affect their choices or preferences of products which can have relevant expressions of environmental concerns, nor even measure their purchase intentions, nor can indicate sustainability impact.
4. Statistical findings and results of the study are not independently tested nor validated; thus, the paper is not scientifically sound as it is.
5. The paper is dangerously misleading as it claims very heavy implications, notions, and generalisations which are not properly supported by data and rigorous sampling of the target population. Refer to section 5 findings and analysis and section 6 discussion and managerial implications.
Thus, it is recommended for the manuscript to be rejected as it is.
Further major revisions can be done to improve the work and may be resubmitted.
Reviewer 2 Report
There are several suggestions which could be useful for author/authors:
Has to be more precise information on sample design and data collection. It seem that data are not representative as it is not ensured that data were collected randomly;
More interpretations of obtained results have to be presented;
More precise indications of novelty/novelties by author/authors as even in the results of the empirical part there are references to other authors;
More precise titles of tables and figures have to be prepared;
Conclusions have to be developed as conclusions and not only as reports on done.
Reviewer 3 Report
This is an interesting study that explores the purchase intentions of those labelled “Gen Z” consumers, with a particular focus on their environmental concerns. The study appears to have been carried out diligently and presents useful findings; I think it is likely to be publishable in the journal after some minor amendments have been made.
In my view, the authors should be invited to address a few points prior:
· - The authors should attempt to place more emphasis, throughout the paper, on what is not known in the present literature, and what is novel about their findings. They tend to present their findings as being in line with, or confirming, prior work in a way that makes this seem like a replication study—which is not the case. This applies in particular to the Introduction, the end of the literature review (where a gap should be identified) and—most especially—to the Discussion section.
· - The literature review might also wish to explore the relationship between “confidence”, which seems an important factor for environmentally-conscious consumption, and “familiarity”, which is associated with influencer-marketing strategies. Are these related constructs?
· - The hypotheses are not phrased in a totally intuitive way. Take H1a: “Environmental concern has a positive impact on eco-labelling”. Eco-labelling is defined on lines 181-183 as a “means” for “organisations”. So the hypothesis does not quite make sense. It is obvious from the surrounding context what is meant, but I think we should be really clear in the text what is meant by the various hypotheses.
· - In section 4 we need to know more about the respondents and their recruitment. How was the questionnaire distributed, and to whom? In what country or countries was this done? Etc.